# Efficacy of Tofacitinib in the Treatment of Universal Alopecia Areata and Primary Sjögren Syndrome

Teresa Rodenas-Herranz [1,2], Marta Cebolla-Verdugo [1,3], Carlos Llamas-Segura [1], Ricardo Ruiz-Villaverde [1,2,*] and Maria Teresa Herranz-Marín [1,3]

1  Hospital Universitario San Cecilio, 18016 Granada, Spain
2  Instituto Biosanitario de Granada—Ibs, 18016 Granada, Spain
3  Hospital Universitario Morales Messeguer, 30008 Murcia, Spain
*  Correspondence: ricardo.ruiz.villaverde.sspa@juntadeandalucia.es





Dear Editor: Alopecia areata (AA) is a form of alopecia whose prevalence ranges from 0.1 to 0.2%. Approximately 14–25% of patients will progress to total hair loss on the head and neck (universal) or the entire body (total). It has been associated with various autoimmune pathologies such as insulin-dependent diabetes mellitus, vitiligo, systemic lupus erythematosus or Sjögren's syndrome, being considered a poor prognostic factor for the disease. Alopecia areata has been associated with numerous human leukocyte antigens such as DQ3, DR4, DR11, and DQ7. Genetic mutations in the genes that encode MHC antigens conditioned by environmental factors such as viral infections, vaccines, low levels of vitamin D in the blood or other possible triggers are what condition the follicular protection processes. The diagnosis is currently based on the clinic and the associated dermoscopy where yellow and black points, exclamation mark hairs, broken hairs, short hairs and tapered hairs are anticipated [1].

The pathophysiology is related to the loss of the immunological privilege of the hair follicle, the recruitment of autoreactive CD8+ lymphocytes and the expression of autoantibodies against follicular antigens. Its therapeutic management is a challenge for the dermatologist and other specialists involved in its comorbidities [2]. A series of clinical cases and retrospective studies have been described in which the use of systemic corticosteroids and classic immunosuppressants such as cyclosporine or methotrexate have been demonstrated with varying degrees of success. Until June 2022, there had been no approval by the USDA (US Food and Drug Administration) for the use of the first drug where severe alopecia areata was indicated on its technical sheet [3].

A 45-year-old woman with no personal or family history of interest was referred to the Dermatology and Internal Medicine Services in May 2018 for a double process that was occurring concurrently.

She presented universal alopecia of 1 year of evolution, with sudden onset (complete loss of cephalic hair in 20 days, including eyebrows and eyelashes, SALT100) (Figure 1A–C). She had followed treatment with topical corticosteroids, two cycles of systemic corticosteroids in a descending pattern (prednisone 30 mg), 5% topical minoxidil with discontinuation of treatment due to local irritation, contact immunotherapy with 2% topical difenciprone in combination with ezetimibe 10 mg and simvastatin 40 mg.

On the other hand, she had simultaneously developed dry skin, mouth, and eyes. No photosensitivity, oral or genital ulcers were reported. Occasionally mild arthralgia in the wrists and knees without inflammatory signs were referred. She did not present hematological and renal involvement in the complementary tests carried out in the previous year.

Complementary tests revealed positive ANA speckled pattern 1/640 with normal levels of C3 and C4, antiDNA negative and positive anti Ro/SSA < 240 and anti La/SSB (=24). The remaining autoimmunity study was negative. Schirmer's test was <5 mm in both

eyes, and salivary gland biopsy showed a local lymphocytic infiltrate but >1 focus/4 mm$^2$ without granulomas with mild acinar atrophy and mild interstitial fibrosis.

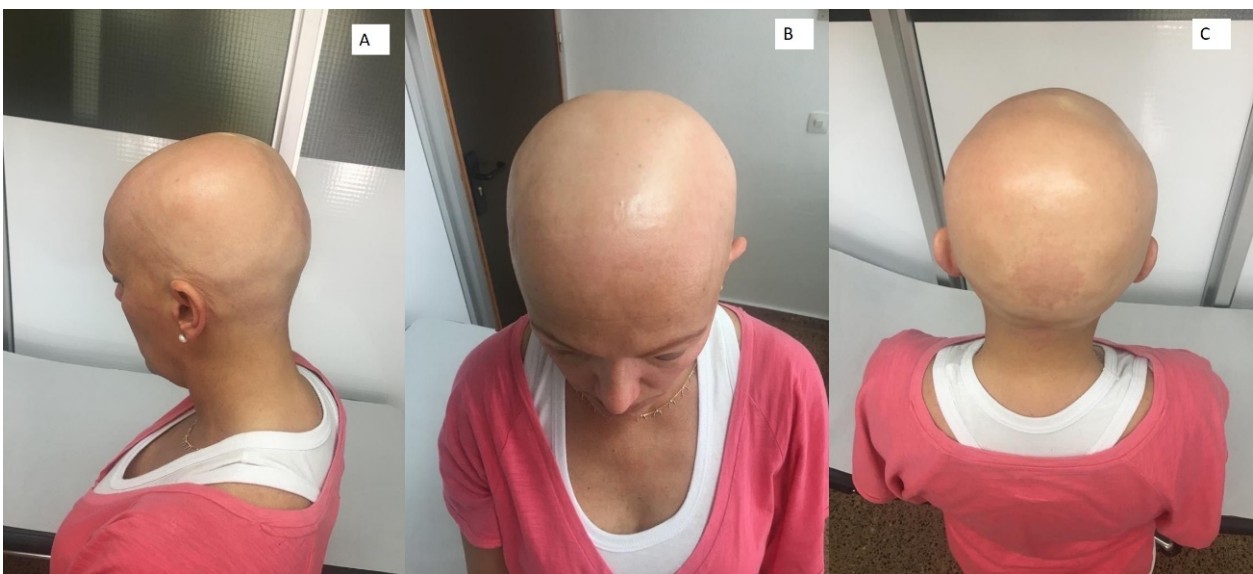

**Figure 1.** (**A**–**C**). Baseline status of our patient prior to starting treatment with tofacitinib (Pfizer Inc, New York, NY, USA) 5 mg every 12 h. SALT 100.

A clinical diagnosis of universal alopecia areata was established in relation to primary Sjögren's syndrome. Treatment with hydroxychloroquine 200 mg/day and dexamethasone 5 mg (0.1 mg/kg) was established 2 consecutive days a week with significant gastrointestinal intolerance, which led to the withdrawal of the medication after 6 months of evaluation. Considering the clinical evolution, the possibility of using tofacitinib 5 mg every 12 h for 13 months after the onset of both processes without suspension of hydroxychloroquine was raised.

At the 6-month follow-up visit, the recovery of the cephalic hair was complete (SALT 0; Figure 2A–C). Eighteen months later, the patient continued treatment without relevant side effects or changes in analytical parameters. Xerostomia and conjunctival dryness have improved substantially, which has also allowed the clinical manifestations of Sjögren's Syndrome to be controlled. Schirmer's test revealed 5 mm on both eyes. Although no objective test was applied to measure the xerostomy, the patient indicated a significant improvement in swallowing processes measured by a visual analogue scale (0–10) with a change in the initial tolerance score from 2 (poor tolerance) to 8 (good tolerance).

Alopecia areata is a non-scarring autoimmune alopecia that produces a significant burden for the patient due to its unpredictable course and the fact that there are no standardized therapies. Until the approval by the EMA in 2022 of baricitinib for use in universal AA, all the treatments used have been off label. The decision was based on the results of two randomized phase 3 clinical trials (BRAVE-AA1 and BRAVE-AA2) involving patients with more than 50% hair loss [1–3]. Around 40% of patients who received baricitinib 4 mg tablets had at least 80% scalp covered by hair in contrast to only 6% of the placebo group after 36 weeks. Side effects include upper respiratory and urinary tract infections, acne, increased low-density cholesterol and creatine kinase levels, and herpesvirus infections [4].

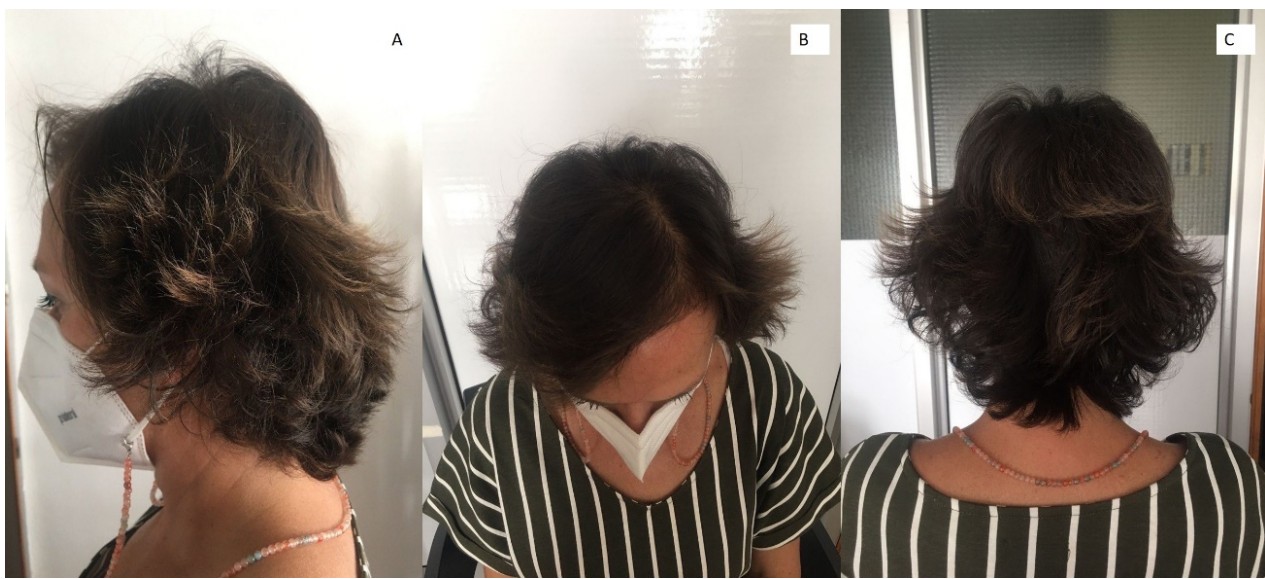

**Figure 2.** (**A–C**) Therapeutic response at 10 months. SALT 0.

Tofacitinib is a JAK inhibitor considered to be panJAK, although its action is exerted preferentially against JAK1 and 3. Blocking the latter may be one of the keys to preventing hair loss and promoting regrowth in patients with AA [5]. In the pathophysiology of AA, JAK induces a blockade of INF-γ and IL-15 signaling in cytotoxic T lymphocytes and keratinocytes of the hair follicle. It is approved by the EMA and FDA for the treatment of rheumatoid arthritis, although it has been used as off-label therapy in the management of other dermatological immune-mediated diseases such as vitiligo, psoriasis or dermatomyositis [6,7].

Different meta-analyses and reviews have been carried out evaluating the use of tofacitinib in AA. The largest includes 275 patients with a repopulation rate of 54% [8]. There are no well-determined predictors of response. The duration of the disease or adherence to treatment have been postulated as factors to consider with unequal response [9]. There do not appear to be differences between the adult and pediatric populations or between the male and female sex. The main side effects that must be monitored include upper respiratory tract infections, gastrointestinal intolerance, hypertension, and other more serious ones such as thrombotic episodes or hematological malignancies.

Regarding Sjögren's syndrome, the JAK-STAT signaling pathway also seems to be involved in its pathogenesis. Tofacitinib may be an effective drug in its control by modulating the autophagy phenomena that occur in the epithelial cells of the salivary glands by decreasing the expression of IL6 that occurs as a consequence of this phenomenon [10]. On the other hand, the epithelial cells of the salivary glands (SGECs) also contribute as protagonists through the production of proinflammatory cytokines such as B lymphocyte activating factor (BAFF) that allows for the survival and maturation of B cells. Interferons type I and II, connectors between innate and adaptive immunity, could induce BAFF expression in SGECs. Its blockade through JAK1 inhibition in the JAK-STAT pathway reduces the inflammatory cascade that perpetuates Sjögren's syndrome. For this reason, drugs such as tofacitinib help to control the symptoms of this autoimmune condition [11,12]. This immunomodulatory effect would also have been studied through the topical application of tofacitinib, both in alopecia areata and in patients with dry eye disease where it could reduce the expression of HLA-DR and proinflammatory cytokines [13,14].

Ruxolitinib selectively inhibits JAK 1 and 2 and to some extent TYK2. In an open clinical trial measuring the efficacy of ruxolitinib in 12 patients with moderate to severe alopecia areata, ruxolitinib was administered at a dose of 20 mg twice daily for 3–6 months with 75% responders with 50% regrowth of the hair that they basally lacked [15]; however, there is no consistent evidence of the use of its topical application.

Data for the efficacy of ritlecitinib in alopecia areata have been from ALLEGRO trials. In a phase 2b/3 randomized, placebo-controlled double-blind trial involving more than 700 patients older than 12 years from all the world, scalp hair loss was required to be greater than 50% persisting for more than 6 months but less than 10 years. After 24 weeks, at least 80% of the patients reached the primary endpoint of marked capillary regrowth [16].

Brepocitinib, a TYK2/JAK1 inhibitor, was developed and tested against ritlecitinib and placebo in a randomized study involving 142 adults with more than 50% hair loss persisting for at least 6 months. At the end of the 24-week period, the proportion of patients who managed to regain at least 70% of their scalp hair was 64% with brebocitinib, 50% with ritlecitinib and only 2% with placebo [16].

The introduction of JAK inhibitors in our therapeutic arsenal will allow us to control dermatological diseases such as AA that were orphaned until now by specific therapies including those that, in cases such as the one presented, can reduce the inflammatory burden of associated autoimmune comorbidities. One of the main points to develop will be the possible relapse after drug withdrawal, since recent studies have shown that 90% of patients lose their repopulated hair again once treatment has ceased, which indicates the need to prolong this treatment over time [17].

**Author Contributions:** Conceptualization M.T.H.-M.; writing—original draft preparation, T.R.-H., R.R.-V., C.L.-S. and M.C.-V.; writing—review and editing, R.R.-V. All authors have read and agreed to the published version of the manuscript.

**Funding:** This research received no external funding.

**Institutional Review Board Statement:** Ethical review and approval were waived for this study because they are not required in our hospital for the publication of singular clinical cases that do not require an intervention other than usual clinical practice.

**Informed Consent Statement:** Written informed consent has been obtained from the patient to publish this paper.

**Data Availability Statement:** Data are available to authors upon reasonable request.

**Conflicts of Interest:** The authors declare no conflict of interest.

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
