# Peer review of "Efficacy of Tofacitinib in the Treatment of Universal Alopecia Areata and Primary Sjögren Syndrome"

_dermato, doi:10.3390/dermato3020009_

Round 1

Reviewer 1 Report

The authors describe a patient with alopecia universalis who also developed Sjogren’s syndrome. Her hair regrew fully on treatment with oral tofacitinib and her Sjogren’s also appeared to improve.

Comments

The response of alopecia areata to treatment with tofacitinib is now well-established and, on its own, a single case report is not remarkable. There has to be something more!

The novelty is provided by the patient’s concurrent Sjogren’s syndrome (SS). I am not aware of any established association between AA and SS and a comment on this should be made. SS can be primary or secondary, the latter related to other autoimmune diseases such as SLE and rheumatoid arthritis. There is a well-established association between AA and SLE and a brief comment on why this patient had primary rather than secondary SS (lack of anti-DNA antibodies?) would be helpful. 

The patient’s xerostomia and conjunctival dryness improved on tofacitinib. Was this confirmed using an objective test?

The discussion on JAK inhibitors in SS needs to be improved in terms of the content, and the ‘flow’ of the argument. The problem is partly one of language in that ‘would’ and ‘could’ are used inappropriately and it is not clear which statements are speculation and which based on evidence, or which are ‘lab-based’ and which are clinical trials. For example, the Huang study (ref 11) was a clinical trial with a substantial number of patients not ‘would have been studied’. The recent RCT including filgotinib by Price et al is not mentioned (https://doi.org/10.1093%2Frheumatology%2Fkeac167) – this failed to show a significant benefit over placebo in SS, casting some doubt over the potential for JAK-I in this disease.

Author Response

The authors describe a patient with alopecia universalis who also developed Sjogren’s syndrome. Her hair regrew fully on treatment with oral tofacitinib and her Sjogren’s also appeared to improve.

Thank you very much for the comments made to our manuscript that we believe will improve its quality.

Comments

• The response of alopecia areata to treatment with tofacitinib is now well-established and, on its own, a single case report is not remarkable. There has to be something more!

AUTHOR REPLY: Thank you very much for your comment. In this case, the association has only been described on one occasion in the literature and for this reason we find it interesting to report this singular association, although with a single reported clinical case we cannot establish causality.

• The novelty is provided by the patient’s concurrent Sjogren’s syndrome (SS). I am not aware of any established association between AA and SS and a comment on this should be made. SS can be primary or secondary, the latter related to other autoimmune diseases such as SLE and rheumatoid arthritis. There is a well-established association between AA and SLE and a brief comment on why this patient had primary rather than secondary SS (lack of anti-DNA antibodies?) would be helpful. 

AUTHOR REPLY: The difference between primary and secondary SS is not defined by the antibody titer but by the fact that it is associated or not with another autoimmune disease such as LE, for this reason and in the absence of a diagnosis of another connective tissue disease, the diagnosis of SS has been established. primary and as such is made explicit in the manuscript.

• The patient’s xerostomia and conjunctival dryness improved on tofacitinib. Was this confirmed using an objective test?

AUTHOR REPLY: Objective tests were measured in the case of tear secretion improvement but not in the case of xerostomia. We provide the measurement of swallowing tolerance by visual analog scale (VAS) with significant improvement in the score.

• The discussion on JAK inhibitors in SS needs to be improved in terms of the content, and the ‘flow’ of the argument. The problem is partly one of language in that ‘would’ and ‘could’ are used inappropriately and it is not clear which statements are speculation and which based on evidence, or which are ‘lab-based’ and which are clinical trials. For example, the Huang study (ref 11) was a clinical trial with a substantial number of patients not ‘would have been studied’. The recent RCT including filgotinib by Price et al is not mentioned (https://doi.org/10.1093%2Frheumatology%2Fkeac167) – this failed to show a significant benefit over placebo in SS, casting some doubt over the potential for JAK-I in this disease.

AUTHOR REPLY: The discussion has been expanded with the data requested by the reviewer and others that we consider to be of special interest.

Reviewer 2 Report

This case report is well-documented and provides valuable insight into the potential use of Tofacitinib for the treatment of universal alopecia areata in patients with primary Sjögren's syndrome,including its pathophysiology, associated autoimmune pathologies.

However, there are some concerns the authors need to address:

1.  The introduction could be expanded upon to provide more background information and context for readers who are not familiar with the disease.

2.  The report concludes that the patient's condition is improved with Tofacitinib treatment, do the authors have further information concerning the patient's long-term prognosis and the possibility of relapse?

3.  Can the authors mention or compare other JAK inhibitors (such as "ruxolitinib" and "baricitinib") in alopecia in the discussion section?

Author Response

This case report is well-documented and provides valuable insight into the potential use of Tofacitinib for the treatment of universal alopecia areata in patients with primary Sjögren's syndrome,including its pathophysiology, associated autoimmune pathologies.

Thank you very much for the comments made to our manuscript that we believe will improve its quality.

However, there are some concerns the authors need to address:

  1. The introduction could be expanded upon to provide more background information and context for readers who are not familiar with the disease.

AUTHOR REPLY: We have expanded the introduction as indicated by the reviewer.

  1. The report concludes that the patient's condition is improved with Tofacitinib treatment, do the authors have further information concerning the patient's long-term prognosis and the possibility of relapse?

AUTHOR REPLY: The discussion has been expanded with the data requested by the reviewer.

  1. Can the authors mention or compare other JAK inhibitors (such as "ruxolitinib" and "baricitinib") in alopecia in the discussion section?

AUTHOR REPLY: The discussion has been expanded with the data requested by the reviewer.